# Job satisfaction as a moderator between organizational culture and turnover intention among Malaysian police officers

Erny Yusnida Che Yusoff🔗[1]*, Mohd Nasir Selamat[2], Rusyda Helma Mohd[2]

1 Centre for Research in Psychology and Human Well-being, Faculty of Social Sciences and Humanities, The National University of Malaysia, Bangi, Selangor, Malaysia, 2 Centre for Research in Psychology and Human Well-being, Faculty of Social Sciences and Humanities, The National University of Malaysia, Bangi, Selangor, Malaysia

* p119402@siswa.ukm.edu.my

## Abstract

Turnover among police officers poses a serious challenge to law enforcement agencies by affecting operational stability and public trust. While organizational culture and job satisfaction are key factors influencing turnover intention, their combined effects have not been widely studied in policing contexts. This study explores whether job satisfaction moderates the relationship between organizational culture and turnover intention among Malaysian police officers. Although both variables were significantly related to turnover intention, the moderation effect of job satisfaction was not statistically significant. A cross-sectional survey was conducted among 373 officers from various departments of the Royal Malaysia Police. Participants completed validated instruments measuring organizational culture, job satisfaction, and turnover intention. Data were analysed using Hayes' PROCESS Macro (Model 1) in SPSS to test for moderation effects through hierarchical regression. The analysis revealed that both organizational culture and job satisfaction were significantly and negatively associated with turnover intention, indicating that higher levels of these variables correspond to lower intentions to leave the organization. Officers who perceived stronger organizational culture and reported higher job satisfaction were less likely to express intent to leave. However, job satisfaction did not significantly moderate the relationship between organizational culture and turnover intention ($p = .117$), indicating that the influence of organizational culture on turnover intention remained stable regardless of satisfaction levels. The findings highlight the importance of fostering a supportive organizational culture to reduce turnover intention among police officers. While job satisfaction independently contributes to retention, it does not significantly alter the impact of organizational culture. These results suggest that efforts to improve retention in policing should focus on structural and cultural reforms in addition to enhancing job satisfaction.

**Data availability statement:** All relevant data are within the paper and its Supporting Information files.

**Funding:** The author(s) received no specific funding for this work.

**Competing interests:** NO authors have competing interests.

## Introduction

Understanding turnover intention among police officers is critical for ensuring the effectiveness, continuity, and public trust in law enforcement institutions. High rates of attrition can disrupt operational readiness, increase training costs, and erode institutional memory, ultimately compromising public safety and the quality of policing services [1,2].

Turnover intention refers to an employee's conscious and deliberate decision to leave the organization. In law enforcement an inherently high-risk and high-stress profession turnover intention is often influenced by factors such as organizational culture, job satisfaction, perceived fairness, and support systems [3,4]. Among these, organizational culture and job satisfaction are consistently recognized as key predictors of employee retention. Organizational culture encompasses the shared values, beliefs, and norms that shape behavior and influence employees' perceptions of leadership, decision-making, and workplace support [2,5]. A positive organizational culture, marked by ethical leadership and inclusivity, is known to reduce stress and increase organizational commitment [6–8].

Job satisfaction, defined as an individual's emotional and cognitive evaluation of their job role, work environment, and career prospects, is another critical factor in employee well-being and retention. In policing, satisfaction is influenced by supervisory support, work-life balance, recognition, and promotional opportunities [1,9,10]. Satisfied officers are more likely to report psychological resilience and lower intention to leave, even when confronted with occupational stressors [11,12]. Several scholars have proposed that job satisfaction not only serves as a direct predictor of employee retention, but may also act as a moderating variable that can amplify or diminish the influence of organizational culture on turnover intention [13]. While past studies have examined direct links between organizational culture, job satisfaction, and turnover intention, few have explored their interactive effects, particularly in the law enforcement context of Malaysia [9,13–15]. Furthermore, limited research has tested the hypothesis that job satisfaction moderates the relationship between organizational culture and turnover intention. This oversight is significant, as Malaysian police officers operate in environments marked by rigid hierarchies, emotional strain, and limited upward mobility factors that may uniquely shape how these variables interact [14,15]. High turnover within the RMP has been linked to perceived inequalities, weak organizational support, and unresolved dissatisfaction with leadership and career progression [14,15]. Moreover, much of the existing literature has been situated in corporate or healthcare sectors, with limited focus on the public security domain [13,15,16]. These issues underscore the urgency of conducting a focused study that explores the interplay between organizational culture, job satisfaction, and turnover intention within this crucial sector.

Recent international research underscores the critical role of supportive organizational cultures in reducing turnover rates among police personnel [17]. These findings are complemented by empirical evidence demonstrating that job satisfaction is significantly influenced by factors such as well-defined job roles, effective supervisory support, clear promotion pathways, and sufficient institutional resources [7]. Despite

these established associations, it remains uncertain whether job satisfaction can effectively buffer the negative impact of a weak organizational culture on officers' intention to leave particularly within the distinctive structural, cultural, and operational environment of Malaysian policing [8].

Therefore, this study aims to examine the relationships between organizational culture, job satisfaction, and turnover intention among Malaysian police officers, with a specific focus on assessing whether job satisfaction moderates the effect of organizational culture on turnover intention. Addressing this gap contributes both theoretically and practically by enhancing the understanding of turnover mechanisms in policing and informing policies designed to improve workforce stability in high-demand public safety sectors.

## Literature review and theoretical framework

Turnover intention among police officers remains a persistent concern in organizational and occupational psychology, particularly in high-stress, hierarchical environments such as law enforcement. While prior studies have explored the impact of organizational culture and job satisfaction on employee retention, the complex interplay between these factors especially the potential moderating role of job satisfaction has received limited empirical attention in policing contexts. Addressing this gap, the present study draws upon relevant theoretical models and empirical findings to examine how job satisfaction might influence the relationship between organizational culture and turnover intention. The following literature review provides a structured synthesis of these components and outlines the conceptual foundation for the study's moderation hypothesis.

## Literature review

This literature review outlines key definitions, synthesizes prior research findings, and highlights theoretical models relevant to the study. The sequence follows a logical structure, beginning with organizational culture, followed by job satisfaction, turnover intention, the proposed moderation hypothesis, and finally, the theoretical framework that underpins the research.

## Organizational culture

Organizational culture refers to the shared values, beliefs, and behavioural norms that guide interactions and expectations within an institution [2,5]. In policing, organizational culture significantly influences performance, cohesion, and officers' psychological well-being. A culture characterized by ethical leadership, transparency, and inclusivity has been associated with stronger organizational commitment and reduced turnover [6–8]. Sulaiman [2] and Ghazali & Asmawi [5] reported that supportive cultural practices are associated with higher job satisfaction and lower turnover intention among Malaysian police officers. Jabeen et al. [6] emphasized that ethical and collaborative cultures enhance officers' sense of belonging and psychological safety, ultimately strengthening organizational commitment. International findings by O'Reilly et al. [7] and Carlan [8] further support that well-established organizational cultures reduce attrition by fostering alignment between personal and institutional values. Recent works by Gomes et al. [18] and Papazoglou & Andersen [19] reaffirm that organizational support structures are crucial in shaping the psychological climate of policing. In contrast, rigid hierarchies and lack of trust can undermine morale and increase attrition, especially in high-demand environments such as law enforcement.

## Job satisfaction

Job satisfaction is defined as an individual's positive or negative evaluative judgment of their job, including aspects such as leadership, recognition, role clarity, and work-life balance [1,9,10]. Recent literature has also emphasized that job satisfaction plays a dual role. It functions both as an outcome of favorable organizational conditions and as a protective factor

 

that buffers employees from workplace stress. This dual function highlights job satisfaction as an indicator of employee well-being and a mechanism that mitigates the negative effects of challenging work environments [11,12]. This aligns with the principles of the Job Demands-Resources (JD-R) Model, where job satisfaction acts as a personal resource that mitigates the adverse impact of workplace stressors. Officers who report high job satisfaction typically experience less psychological strain and exhibit greater resilience, which contributes to lower turnover intention [11,12].

### Turnover intention

Turnover intention is the conscious inclination of an employee to voluntarily leave their organization. In law enforcement, high turnover intention has been linked to organizational injustice, poor leadership, and a lack of advancement opportunities [3,4]. Strengthening organizational culture and enhancing job satisfaction are key strategies for reducing attrition [20]. This view is reinforced by Van Gelderen et al. [4], who found that burnout and lack of support systems significantly increase turnover among police personnel.

### Moderation Hypothesis: Job Satisfaction as a Buffer

Emerging research suggests that job satisfaction may act as a moderating variable, capable of attenuating the negative effects of a poor organizational culture on turnover intention [14–17]. Satisfied officers are less likely to act on negative workplace perceptions [14], maintain their emotional commitment and reduce the desire to exit [17]. These results suggest that job satisfaction not only predicts turnover intention directly but also modifies the strength of other contributing factors, particularly organizational culture. Additional support comes from Shane [21] and Trinkner et al. [22], who documented how satisfaction-based interventions reduce voluntary exits in law enforcement. Furthermore, Anshel et al. [23] and He et al. [24] point out that resilience and coping resources interact with satisfaction levels to protect against the harmful effects of organizational dysfunction. However, empirical support for this moderation effect remains limited and inconsistent, particularly in the Malaysian policing context, where institutional norms, occupational stressors, and hierarchical structures may constrain the buffering role of job satisfaction [8].

### Theoretical framework

The theoretical framework provides a clearer conceptual basis for the moderation hypothesis. Specifically, the study in two complementary theoretical models that justify the examination of job satisfaction as a moderator. This study is grounded in two theoretical models. First, Social Exchange Theory [25] posits that employees who perceive strong organizational support tend to reciprocate with increased loyalty and reduced turnover intention. In policing, where trust, reciprocity, and mutual respect are paramount a supportive organizational culture fosters stronger retention. This is reinforced by empirical studies such as Francis et al. [9] and Chong & Mui [26] validate the relevance of this theory among police officers.

Second, Job Demands-Resources (JD-R) Model [27] provides a framework for understanding how organizational and personal resources interact to mitigate job stress. Within this model, organizational culture is conceptualized as a job resource, while job satisfaction serves as a personal resource that may moderate the impact of stressors on outcomes such as turnover intention [12,18]. Together, these models offer a theoretical basis for examining whether job satisfaction influences the strength of the relationship between organizational culture and turnover intention. The dual role of job satisfaction captures both its intrinsic value and its interactional influence.

### Research hypotheses

Building on the preceding literature, this study formulates three hypotheses grounded in the Social Exchange Theory [25] and the Job Demands-Resources (JD-R) Model [28]. These theoretical perspectives posit that organizational culture and job satisfaction are central to shaping employees' behavioral intentions, particularly their intention to remain with or

 

leave the organization. Within the high-stress, hierarchical environment of the Royal Malaysia Police, understanding these dynamics is critical for improving retention strategies.

**H1: Organizational culture is significantly related to turnover intention among Malaysian police officers.**

Organizational culture encompasses the shared values, norms, and practices that shape behavior and attitudes within a workplace. A supportive culture marked by ethical leadership, fairness, and inclusivity has been found to reduce employees intention to leave [6,7]. According to Social Exchange Theory, when officers perceive that their organization treats them with respect and fairness, they are more likely to reciprocate with loyalty and reduced turnover intentions [25,28]. In policing, where organizational trust is essential, a positive culture plays a vital role in enhancing commitment and retention.

**H2: Job satisfaction is significantly related to turnover intention among Malaysian police officers.**

Job satisfaction refers to an individual's overall contentment with their job, encompassing factors such as career development, supervisory support, and work-life balance. According to the Social Exchange Theory, satisfied employees are more likely to view their organization as fulfilling its obligations, thereby fostering a sense of obligation to remain [25]. Empirical studies by Lambert et al. [29], Carlan [8], and Garcia et al. [30] demonstrate a consistent negative relationship between job satisfaction and turnover intention, particularly in policing contexts where dissatisfaction often arises from bureaucratic constraints or lack of recognition. In the Royal Malaysia Police, officers with higher satisfaction levels may exhibit stronger commitment and reduced desire to exit the force.

**H3: Job satisfaction significantly moderates the relationship between organizational culture and turnover intention among Malaysian police officers, such that the relationship is weaker at higher levels of job satisfaction.**

Drawing on the JD-R Model, organizational culture is conceptualized as a job resource, while job satisfaction acts as a personal resource that can buffer the impact of workplace stressors. Prior research indicates that job satisfaction can attenuate the effects of negative organizational conditions, making it a potential moderator [14,17]. When officers experience a high level of satisfaction, the adverse effects of a rigid or unsupportive culture on their turnover intention may be reduced. Conversely, in the absence of satisfaction, even moderately positive organizational cultures may not retain officers. Thus, this study tests whether job satisfaction interacts with organizational culture to influence turnover intention.

The conceptual framework of the study is illustrated in Fig 1.

## Materials and methods

This study adopted a quantitative, cross-sectional survey design to investigate the relationships among organizational culture, job satisfaction, and turnover intention. The design also enabled the testing of a moderation effect, with job satisfaction hypothesized to moderate the relationship between organizational culture and turnover intention.

### Sampling procedure

A stratified random sampling technique was employed to ensure a representative distribution of police officers across various divisions, ranks, and departments within the Royal Malaysia Police (RMP). This approach allowed proportionate sampling from different departments and geographic regions, thereby enhancing the generalizability of the findings across

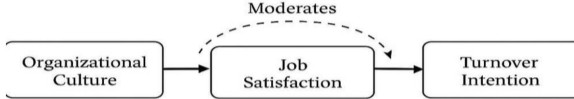

**Fig 1. The conceptual framework of the study.**

the RMP. Participants were recruited between 25/07/2024 until 08/08/2024. During this period, a total of 500 structured questionnaires were distributed through official internal communication platforms, and 373 were returned and deemed usable, yielding a response rate of 74.6%. The data were collected anonymously to preserve respondent confidentiality. Participation was voluntary, and informed written consent was obtained from all participants prior to data collection. This study received approval from the Human Research Ethics Committee of The National University of Malaysia (Ref No: JEP-2024–230). No minors were involved in this study, and confidentiality was assured throughout the research process.

## Survey instrument

Data were collected using a structured self-administered questionnaire consisting of four sections: demographic profile, organizational culture, job satisfaction, and turnover intention. The instruments were adapted from validated scales:

Organizational Culture: Based on the Organizational Culture Assessment Instrument (OCAI).

Job Satisfaction: Adapted from the Job Satisfaction Survey (JSS).

Turnover Intention: Measured using items developed by Mobley et al.

All items were rated on a 7-point Likert scale (1 = strongly disagree to 7 = strongly agree).

## Instrument validation

Prior to full deployment, a pilot study involving 30 police officers was conducted to test item clarity and reliability. Minor adjustments were made for clarity and cultural appropriateness: Organizational Culture: $\alpha = 0.87$; Job Satisfaction: $\alpha = 0.91$; Turnover Intention: $\alpha = 0.89$. Exploratory Factor Analysis (EFA) confirmed construct validity.

## Assumption testing

Several statistical assumptions were assessed prior to regression analysis: Normality was assessed using skewness and kurtosis values, which were within the acceptable range (±1); Multicollinearity was examined through Variance Inflation Factor (VIF) with all values below 5; Homoscedasticity and linearity were confirmed through residual scatterplots.

## Data analysis

All statistical analyses were performed using IBM SPSS Statistics version 26. Preliminary descriptive statistics were used to summarize demographic characteristics of the respondents (as presented in Table 1) and to examine item distributions across the main study variables. Frequencies and percentages were computed for categorical variables such as gender, age, rank, and department to establish the representativeness of the sample within the Royal Malaysia Police.

Following the descriptive analysis, Pearson correlation analysis was conducted to assess bivariate relationships among the key continuous variables organizational culture, job satisfaction, and turnover intention. These correlations provided initial insights into the direction and strength of the associations.

To test the moderation hypothesis, a hierarchical multiple regression analysis was performed using Hayes' PROCESS Macro (Model 1). This method was chosen for its ability to estimate interaction effects and conditional relationships across levels of the moderating variable (job satisfaction). The hierarchical approach allowed for the assessment of incremental variance explained by the interaction term (organizational culture × job satisfaction) above and beyond the individual main effects.

The analysis also employed bootstrapping procedures with 5,000 resamples to generate robust estimates and confidence intervals, thereby enhancing the reliability of the moderation findings. Although the interaction term was not statistically significant, further interpretation was supported by simple slopes analysis and visual inspection of the interaction plot.

As illustrated in Fig 2, the conditional effect of organizational culture on turnover intention was examined at three levels of job satisfaction: low (4.77), moderate (5.58), and high (6.39). The plot revealed a counterintuitive trend where turnover

**Table 1. Demographic profile of the respondents.**

| Demographic Variable | Category | Frequency (n) | Percentage (%) |
|---|---|---|---|
| **Gender** | Male | 262 | 70.2 |
| | Female | 111 | 29.8 |
| **Age Group** | 21–30 years | 119 | 31.9 |
| | 31–40 years | 169 | 45.3 |
| | 41–50 years | 60 | 16.1 |
| | Above 50 years | 25 | 6.7 |
| **Rank** | Inspector and below | 247 | 66.2 |
| | ASP and above | 126 | 33.8 |
| **Department** | Criminal Investigation | 92 | 24.7 |
| | Narcotics | 85 | 22.8 |
| | Traffic | 76 | 20.4 |
| | General Administration/Other | 120 | 32.1 |

Note: ASP = Assistant Superintendent of Police.

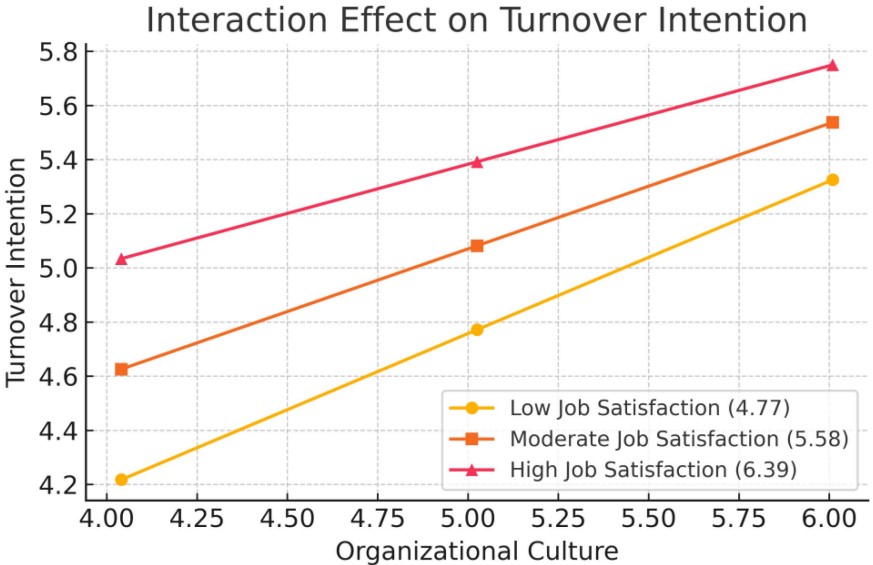

**Fig 2. Interaction effect of organizational culture and job satisfaction on turnover intention at low, moderate, and high levels of job satisfaction.**

intention increased with stronger perceptions of organizational culture, particularly among those with high job satisfaction. This anomaly suggests that job satisfaction may not uniformly buffer the influence of organizational culture as expected and may be shaped by unmeasured contextual or psychological factors.

To complement the analysis, Table 2 presents the results of the hierarchical regression models, including regression coefficients, R-squared changes, and significance levels for both main and interaction effects. This integrated analytical strategy provides a rigorous foundation for evaluating the interplay between organizational culture and job satisfaction in predicting turnover intention and informs the practical recommendations presented in the discussion section.

**Table 2. Effect of Organizational Culture on Turnover Intention with Job Satisfaction as a Moderator.**

| Predictor | B | SE | t | p | LLCI | ULCI |
|---|---|---|---|---|---|---|
| Constant | −2.8346 | 2.2722 | −1.2476 | .2130 | −7.3026 | 1.6334 |
| Organizational Culture | 1.1495 | 0.4546 | 2.5286 | .0119 | 0.2556 | 2.0435 |
| Job Satisfaction | 1.0021 | 0.4050 | 2.4744 | .0138 | 0.2057 | 1.7985 |
| Organizational Culture X Job Satisfaction | −1.1231 | 0.7083 | −1.5712 | .1170 | −2.7711 | 0.0310 |

Constant: Turnover Intention.

## Results

This chapter presents the results of the statistical analyses conducted to examine the relationships among organizational culture, job satisfaction, and turnover intention, as well as the moderating effect of job satisfaction on the relationship between organizational culture and turnover intention among Malaysian police officers.

### Descriptive statistics

To provide a comprehensive overview of the sample, descriptive statistics were computed to summarize the demographic characteristics of the 373 police officers involved in the study, alongside measures of central tendency and dispersion for the main study variables. This section presents the demographic profile of respondents by gender, age, rank, and departmental affiliation to illustrate the diversity and representativeness of the sample. Additionally, mean scores and standard deviations for organizational culture, job satisfaction, and turnover intention are reported to capture general trends in perceptions and attitudes among police personnel. These descriptive insights lay the groundwork for further inferential analysis, enabling an informed interpretation of subsequent correlation and regression results.

### Demographic characteristics

Table 1 Summarizes the demographic characteristics of the 373 police officers who participated in the study. In terms of gender, the sample comprised 262 males (70.2%) and 111 females (29.8%), reflecting the male-dominated composition typical of law enforcement agencies in Malaysia.

Regarding age distribution, most respondents (45.3%) were in the 31–40 year age group, followed by those aged 21–30 years (31.9%). Officers aged 41–50 years accounted for 16.1% of the sample, while only 6.7% were aged above 50. This indicates a workforce predominantly in the early and mid-career stages.

In terms of rank, 66.2% of respondents held positions of Inspector or below, while 33.8% were Assistant Superintendents of Police (ASP) or above, demonstrating a reasonably balanced representation of lower and upper ranks in the Royal Malaysia Police (RMP).

The distribution across departments was relatively diverse. Officers from the General Administration or other units made up the largest group (32.1%), followed by the Criminal Investigation Department (24.7%), Narcotics Division (22.8%), and Traffic Division (20.4%). This diversity allows for broader generalization of the findings across operational and administrative branches within the police force.

Descriptive analysis revealed the following mean scores and standard deviations for the main study variables:

- Organizational Culture: M = 5.12, SD = 0.76

- Job Satisfaction: M = 5.58, SD = 0.81

- Turnover Intention: M = 3.79, SD = 0.93

These results indicate that, on average, respondents reported moderately high levels of perceived organizational culture and job satisfaction, while turnover intention was relatively lower. The standard deviations suggest a moderate spread of responses, indicating some variability in how these constructs were experienced among police officers.

## Correlation analysis

To examine the relationships between the key variables, Pearson correlation analysis was conducted. The results demonstrated statistically significant associations in the expected directions:

- Organizational culture was negatively correlated with turnover intention (r = −0.31, p < 0.01) indicating that more positive perceptions of organizational culture were associated with lower levels of intention to leave.

- Job satisfaction was negatively correlated with turnover intention (r = −0.45, p < 0.01) suggesting that officers who reported greater satisfaction with their work were significantly less likely to consider leaving the organization.

- Organizational culture was positively correlated with job satisfaction (r = 0.52, p < 0.01) highlighting that supportive and values-driven organizational environments tend to promote higher employee satisfaction.

These findings offer preliminary support for the hypothesized model, in which organizational culture is proposed to influence turnover intention both directly and indirectly through job satisfaction. The statistically significant correlations also reinforce the theoretical assumptions drawn from the Social Exchange Theory and the Job Demands-Resources (JD-R) Model, providing a foundational basis for subsequent regression and moderation analyses.

## Hierarchical regression analysis and moderation test

A hierarchical regression analysis using Hayes' PROCESS Macro (Model 1) was performed to test the moderating effect of job satisfaction. The analysis was conducted in three steps:

- Model 1: Entered organizational culture and job satisfaction as predictors.

- Model 2: Added the interaction term (Organizational Culture × Job Satisfaction).

The overall model was statistically significant in predicting turnover intention. However, the interaction term was not significant, indicating that job satisfaction does not significantly moderate the relationship between organizational culture and turnover intention.

## Effect of organizational culture on turnover intention with job satisfaction as a moderator

As shown in Table 2, both organizational culture and job satisfaction have a significant positive effect on turnover intention independently, with p values of 0.0119 and 0.0138, respectively. However, Table 2 displays the interaction effect of job satisfaction as a moderator, which was found to be non-significant (p = 0.1170). These result suggest that while each variable contributes to explaining turnover intention, job satisfaction does not significantly moderate the relationship between organizational culture and turnover intention.

The main effect of organizational culture on turnover intention was significant, b = 1.1495, SE = 0.4546, t = 2.5286, p = 0.0119, 95% CI 0.2556, 2.0435. This suggests that higher levels of organizational culture are associated with increased turnover intention. Similarly, the main effect of job satisfaction was also significant, b = 1.0021, SE = 0.4050, t = 2.4744, p = 0.0138, 95% CI 0.2057, 1.7985. This indicates that higher levels of job satisfaction are related to increased turnover intention, which may reflect specific dynamics within this sample of police officers.

The interaction effect between organizational culture and job satisfaction was not statistically significant (b = −1.1231, SE = 0.0783, t = −1.5712, p = 0.1170, 95% CI [−0.2771, 0.0310]). However, both organizational culture (b = 1.1495,

p = 0.0119) and job satisfaction (b = 1.0021, p = 0.0138) were significant positive predictors of turnover intention. This outcome aligns with the plotted trend, which illustrates that as either organizational culture or job satisfaction increases, so too does turnover intention. Notably, the negative coefficient of the interaction term suggests a tendency for the joint effect of high organizational culture and high job satisfaction to slightly reduce turnover intention, although this interaction was not strong enough to reach statistical significance. Therefore, the findings support the conclusion that job satisfaction does not significantly moderate the relationship between organizational culture and turnover intention.

The model summary indicates a moderate correlation (R = 0.4814) and shows that approximately 23.18% of the variance in turnover intention ($R^2$ = 0.2318) can be explained by the predictors in the model, with a mean square error (MSE) of 1.4628. The model as a whole is statistically significant (F(3, 369) = 37.1098, p < .001), suggesting that the predictors significantly contribute to explaining turnover intention. However, the interaction effect between organizational culture and job satisfaction on turnover intention is not statistically significant, as indicated by a negligible change in $R^2$ ($\Delta R^2$ = 0.0051) and a non-significant F(1, 369) = 2.4687, p = 0.1170. This implies that job satisfaction does not significantly moderate the relationship between organizational culture and turnover intention in this model.

### Interaction term analysis showing moderation effect

Table 3 presents the findings indicate that both organizational culture and job satisfaction are significant predictors of turnover intention independently. However, job satisfaction does not significantly moderate the relationship between organizational culture and turnover intention in this sample of Malaysian police officers. Further research could explore additional factors that might influence this relationship to provide more comprehensive insights into turnover intentions within police forces.

### Interpretation of moderation plot

**Interaction effect of organizational culture and job satisfaction on turnover intention at low, moderate, and high levels of job satisfaction.** Fig 2. Presents the interaction between organizational culture and job satisfaction in predicting turnover intention at three levels of job satisfaction: low (M − 1 SD = 4.77), moderate (M = 5.58), and high (M + 1 SD = 6.39). The interaction effect was not statistically significant (b = −1.1231, SE = 0.0783, p = .1170), indicating that job satisfaction did not moderate the relationship between organizational culture and turnover intention. However, when visually inspecting the interaction plot, a pattern emerges in the slopes across job satisfaction levels. Specifically, at low levels of job satisfaction, turnover intention remains relatively unchanged as organizational culture varies, as reflected by the flatter slope. At moderate and high levels of job satisfaction, the positive slope suggests that turnover intention somewhat increases with stronger perceptions of organizational culture among these groups. Notably, the steepest slope appears for those with high job satisfaction, although this difference does not reach statistical significance. Therefore, while a conditional effect is visually suggested, it is not supported by statistical evidence in this sample.

At first glance, the finding that higher organizational culture appears to coexist with higher turnover intention particularly among those with greater job satisfaction seems counterintuitive, as traditional perspectives suggest a supportive culture should reduce the desire to leave. Within the context of this study among Malaysian police officers, this pattern may reflect a potential misalignment between officers' personal expectations and organizational cultural values. Officers with high

**Table 3. Interaction term analysis showing moderation effect.**

| Interaction Term | R² Change | F | df1 | df2 | p |
|---|---|---|---|---|---|
| Organizational Culture X Job Satisfaction | .0051 | 2.4687 | 1 | 369 | .1170 |

Constant: Turnover Intention.

job satisfaction could also have elevated expectations of the organization; if the organizational culture, even when perceived as strong, does not align with these expectations, it may lead to dissatisfaction or disengagement. Consequently, officers with high satisfaction but low cultural congruence may still express intent to leave. This dynamic is consistent with person-organization fit theory, which posits that mismatches between individual and organizational values can drive turnover intentions even when overall satisfaction is high.

## Summary

The results revealed significant relationships among the key study variables. Both organizational culture and job satisfaction were negatively correlated with turnover intention, while organizational culture was positively correlated with job satisfaction, supporting the hypothesized model. Hierarchical regression analysis further demonstrated that organizational culture and job satisfaction significantly predicted turnover intention independently. However, the interaction term between organizational culture and job satisfaction was not statistically significant, indicating that job satisfaction did not moderate the relationship.

Although the moderation effect was not supported statistically, visual inspection of the interaction plot suggested a counterintuitive trend officers with higher job satisfaction exhibited stronger turnover intention as perceptions of organizational culture improved. This pattern may reflect a mismatch between personal expectations and perceived cultural values within the organization.

The overall model explained 23.18% of the variance in turnover intention, confirming that both predictors are meaningful in understanding turnover behaviour among Malaysian police officers. However, the absence of a significant moderating effect implies that job satisfaction alone may not buffer the influence of organizational culture. These findings highlight the complexity of turnover dynamics and underscore the need to explore additional psychological and organizational variables in future studies.

## Discussion

This study set out to examine whether job satisfaction moderates the relationship between organizational culture and turnover intention among Malaysian police officers. While both organizational culture and job satisfaction independently predicted turnover intention, the anticipated moderating effect of job satisfaction on the relationship between organizational culture and turnover intention was not statistically significant. Nevertheless, the graphical representation suggested notable patterns that merit further discussion.

Interestingly, the moderation analysis revealed a direction of interaction that was not entirely in line with theoretical expectations. While previous literature typically supports the notion that a positive organizational culture reduces turnover intention [31], this study found that even in the presence of high job satisfaction, the influence of organizational culture on turnover intention remained weak or counterintuitive. One plausible explanation lies in the paramilitary and hierarchical nature of police institutions, where cultural norms such as rigid authority, limited upward communication, or bureaucratic constraints may be perceived as misaligned with individual officers' values. Thus, even satisfied officers may still harbor intentions to leave due to structural or cultural dissonance.

Moreover, in high-stress occupations such as policing, job satisfaction may be sustained by personal or interpersonal factor such as camaraderie, field autonomy, or extrinsic rewards rather than alignment with organizational culture per se. This could explain why job satisfaction, although high, does not significantly buffer the effects of perceived cultural rigidity. These findings are consistent with recent studies by [6,15] which found that in public security organizations, retention is more strongly driven by perceptions of procedural fairness and psychological safety than by cultural cohesion alone. Therefore, the counterintuitive directionality may reflect deeper tensions between institutional demands and individual meaning-making within the unique sociocultural setting of the Royal Malaysia Police.

The coexistence of high job satisfaction with high turnover intention, although seemingly paradoxical, may be understood through a more contextualized lens. In hierarchical and high-pressure environments like policing, job satisfaction

may stem from factors such as camaraderie, a sense of purpose, and job security yet these same officers may still contemplate leaving due to other structural or psychological pressures. For instance, officers may feel emotionally exhausted, perceive limited opportunities for upward mobility, or struggle with work-life balance, despite being otherwise satisfied with their current roles.

Additionally, satisfied employees may still consider external opportunities that offer better career prospects, remuneration, or more flexible working conditions, especially in contexts where rigid institutional norms limit personal growth. This pattern aligns with the dual-factor perspective discussed by [32], in which motivators (e.g., recognition, meaningful work) can coexist with hygiene factors (e.g., bureaucracy, job stress) that still prompt withdrawal cognitions. Recent findings by Jabeen et al. [6] and Memory & Emeti [33] also support this complexity, noting that satisfaction is not always a sufficient deterrent to turnover when systemic frustrations remain unresolved.

To further clarify these findings, we provide a more detailed interpretation grounded in theory and context:

1. Multidimensionality of Job Satisfaction: Job satisfaction is not a monolithic construct. Officers may report high satisfaction in certain job facets such as peer relationships, job stability, or pay but still experience dissatisfaction with other aspects, such as leadership support, organizational fairness, or career progression. This fragmented satisfaction may explain why overall job satisfaction coexists with high turnover intention. This perspective is supported by Carlan [8] and Van Gelderen et al. [4]

2. Career Aspirations and Mobility: Police officers who are highly satisfied in their roles may still express intention to leave if they perceive limited opportunities for advancement or professional growth. In such cases, turnover intention is not driven by dissatisfaction per se, but rather by unmet aspirations or ambition. This aligns with findings from McElroy et al. [34], who noted that satisfaction can exist alongside intent to leave when career stagnation is present.

3. Misalignment of Values and Organizational Culture: Officers who value innovation, autonomy, or modern policing approaches may experience a mismatch with a rigid, hierarchical organizational culture. Even if they are otherwise satisfied with their job tasks, such misalignment can generate psychological discomfort, prompting a desire to exit. This is consistent with Person-Organization Fit Theory, which suggests that cultural congruence plays a critical role in employee retention.

4. Implications for Interpretation: These findings suggest that job satisfaction alone may not be sufficient to retain officers unless it is supported by cultural alignment, fairness, leadership support, and opportunities for growth. This highlights the need for holistic retention strategies that go beyond surface-level satisfaction.

## Comparison with previous literature

The positive association between organizational culture and turnover intention may seem counterintuitive, as a strong organizational culture is generally assumed to promote employee retention [7]. However, within the Malaysian policing context, a highly structured and hierarchical culture may be perceived as restrictive or unresponsive to individual needs, especially by officers with higher expectations or autonomy-driven values. This observation aligns with findings from Carlan [8] who reported that formalized and bureaucratic police structures may inadvertently contribute to job dissatisfaction and increased turnover intention, particularly among younger or more progressive officers. Johnson et al. [12] and Garcia et al. [30] supported further by highlighting that police cultures may amplify stress rather than mitigate it, especially in environments lacking psychological support or recognition.

Similarly, the significant positive relationship between job satisfaction and turnover intention requires nuanced interpretation. One possible explanation is that satisfied officers may still consider leaving if they perceive limited career advancement opportunities or misalignment between personal aspirations and institutional values a phenomenon also noted in

McElroy et al. [28] longitudinal analysis of career stagnation in policing. In such cases, satisfaction with one's current role does not necessarily equate to long-term organizational commitment. Carlan [8] and Van Gelderen et al. [4] similarly found that police officers may experience satisfaction with certain job elements (e.g., peer relationships, job security) yet remain disillusioned due to unsupportive leadership or rigid institutional policies. Thus, job satisfaction in policing appears to be multifaceted and context-specific, where satisfaction with micro-level aspects may not overcome the strain caused by macro-level cultural or structural challenges.

### Explaining the non-significant moderation effect

The non-significant interaction effect suggests that job satisfaction does not meaningfully buffer the relationship between organizational culture and turnover intention. This finding contrasts with expectations grounded in Social Exchange Theory and the Job Demands-Resources Model, which predict that personal resources such as job satisfaction should moderate the impact of organizational-level stressors. Several explanations may account for this discrepancy. First, the moderating influence of job satisfaction may be diminished in environments where organizational culture is perceived as rigid or misaligned with modern policing values. Second, other unmeasured variables such as perceived organizational justice, emotional exhaustion, or external career opportunities may mediate or confound this relationship, diluting the moderating role of satisfaction.

The inherently demanding nature of policing, which includes frequent exposure to trauma, high levels of occupational stress, public scrutiny, and rigid institutional hierarchies limit autonomy [12,18]. These organizational realities can diminish the protective influence of individual job satisfaction because broader structural and cultural factors exert a stronger influence on officers' turnover intentions.

Even highly satisfied officers might consider leaving if they perceive a misalignment between their personal values and organizational norms or experience inadequate support from leadership [9]. This finding aligns with literature indicating that in law enforcement contexts, organizational culture variables such as fairness, trust, and leadership integrity may outweigh the effect of job satisfaction on retention decisions [22].

As Lambert et al. [29] and McCreary et al. [11] suggest, additional psychosocial and organizational supports such as access to mental health services, formal mentoring systems, resilience training, and work-life balance programs may play a more influential role in mitigating turnover intentions compared to job satisfaction alone. Recent research emphasizes that perceived organizational support, procedural justice, and opportunities for professional development serve as critical factors in reducing turnover intention beyond the effects of job satisfaction [21,35]. These findings highlight the need for comprehensive retention strategies that go beyond addressing individual satisfaction and instead target systemic organizational improvements.

### Theoretical implications

From a theoretical perspective, the findings challenge the buffering assumption proposed by the Job Demands-Resources (JD-R) Model [27]. According to this model, personal resources such as job satisfaction are expected to mitigate the adverse effects of high job demands or negative organizational conditions, thereby reducing turnover intention. However, the present analysis revealed that job satisfaction did not significantly moderate the relationship between organizational culture and turnover intention among Malaysian police officers. This is consistent with recent studies that suggest the buffering role of job satisfaction may weaken in highly stressful and rigid organizational contexts such as policing [12,18]. These contexts are often characterized by chronic stress, limited autonomy, and rigid hierarchies, which may overshadow the positive effects of personal resources like job satisfaction.

Similarly, while the Social Exchange Theory posits that reciprocal positive treatment by the organization leads to employee commitment and lower turnover intention [25], this mechanism may not operate effectively in policing

environments where perceived organizational support is weak or inconsistent [9,22]. The analysis indicated that even when officers reported relatively high job satisfaction, turnover intentions remained elevated when organizational culture was perceived as unfair or lacking in trust and support. This underscores the dominant role of organizational culture itself as a predictor of turnover intention in policing, rather than job satisfaction acting as a sufficient buffer.

These findings imply that retention strategies in policing institutions should address not only individual-level factors such as job satisfaction, but also broader organizational elements. Strengthening ethical leadership, promoting fairness and inclusivity, and ensuring transparent career progression pathways may yield greater impact in reducing turnover intention among police officers [21,35].

### Practical implications for police management

Practically, these findings highlight the need for police management to critically examine not only the strength but the quality and adaptability of organizational culture. A culture that is strong but inflexible may alienate officers who seek innovation, recognition, or developmental pathways. While enhancing job satisfaction remains important, it is insufficient as a standalone strategy in mitigating turnover intention among police officers, particularly given the unique stressors inherent in policing such as rigid hierarchy, operational risks, and public scrutiny [12,18]. Efforts to improve retention must therefore incorporate broader organizational reforms that target the systemic roots of dissatisfaction and disengagement.

Key areas of focus should include enhancing leadership transparency, promoting an ethical and inclusive organizational culture, ensuring fairness and equity in promotion and disciplinary processes, and providing access to comprehensive mental health and resilience resources [21,35]. Empirical studies have shown that ethical leadership and perceptions of organizational justice are strong predictors of commitment and retention, often outweighing the influence of individual job satisfaction alone [9,22].

Additionally, organizational interventions that foster psychological safety, such as mentoring programs, peer support systems, and opportunities for career development can enhance officers' sense of belonging and reduce the alienation that contributes to turnover [29]. Tailored interventions that address both individual well-being and the structural-cultural transformation of police organizations may therefore be more effective in sustaining workforce engagement and reducing turnover intention in this high-stress, high-risk occupation. Simultaneously, job satisfaction initiatives such as improving supervisory support, recognition systems, and transparent career advancement remain essential, but should be paired with broader organizational reforms to maximize impact.

By adopting a holistic retention strategy grounded in both employee-centred and organization-centered reforms, police departments can not only improve retention rates but also foster a healthier, more resilient, and more effective workforce capable of meeting the complex demands of modern law enforcement.

### Contribution to the field

This study contributes to the literature by expanding the scope of turnover research into the Malaysian law enforcement context, which remains underrepresented in organizational behaviour research. It also challenges the assumption that job satisfaction always moderates organizational influences, suggesting instead that in rigid institutional environments, personal resources may not be sufficient to counteract systemic pressures.

By focusing on the unique occupational and organizational context of the Malaysian police force, this research advances theoretical understanding of workforce dynamics in law enforcement, offering empirical evidence that organizational culture exerts a stronger influence on turnover intention than job satisfaction alone. These findings reinforce the importance of adopting multidimensional retention strategies that target not only individual attitudes but also systemic organizational reforms, including improvements in leadership practices, fairness, and psychological support structures.

Future research should build on these insights by employing longitudinal designs to assess causality over time, integrating qualitative methodologies to capture officers lived experiences in greater depth, and exploring additional

moderating variables such as perceived organizational support, leadership style, occupational identity, and organizational justice. Such efforts will help to clarify the complex mechanisms that underlie turnover intention in policing and provide a more comprehensive evidence base for developing targeted, effective interventions aimed at improving retention and well-being among police officers.

## Limitations, recommendations, and conclusions

Having presented and discussed the key findings of this study, it is important to critically reflect on its limitations, draw out practical recommendations, and outline directions for future research. These final sections aim to contextualize the results within broader theoretical and practical frameworks, highlight areas for improvement in organizational practice, and identify opportunities for advancing empirical knowledge on turnover intention among police officers. Through this, the study offers not only immediate implications for policy and leadership but also sets the stage for deeper investigation into the complex interplay between organizational culture, job satisfaction, and employee retention in law enforcement settings.

### Limitations and future research directions

While this study advances the understanding of the relationship between organizational culture, job satisfaction, and turnover intention among police officers in Malaysia, several limitations should be acknowledged. First, the cross-sectional research design limits the ability to draw causal inferences. Future studies should employ longitudinal designs to examine changes over time and establish causality between these constructs, particularly given the evolving nature of organizational culture and individual job attitudes in policing contexts.

Second, the reliance on self-reported survey data raises the possibility of response bias, especially within a hierarchical and disciplined organization such as the police force, where social desirability pressures may influence responses. Incorporating qualitative methods such as semi-structured interviews or focus group discussions could enrich future research by providing deeper insights into officers lived experiences and the nuanced ways in which organizational culture shapes turnover intentions.

Third, the finding that job satisfaction did not significantly moderate the relationship between organizational culture and turnover intention suggests that alternative variables may better explain this interaction. Future research could explore the moderating or mediating roles of perceived organizational support, psychological resilience, occupational stress coping mechanisms, and organizational commitment [19,23]. These factors have been shown to exert substantial influence on turnover intention in policing and other high-stress occupations [12,18].

Additionally, future studies could examine demographic variations such as rank, gender, and years of service to understand how these characteristics shape officers perceptions of organizational culture and job satisfaction [24]. Such work would help to identify subgroups that may be more vulnerable to disengagement and turnover, thereby enabling the design of targeted retention strategies.

### Practical recommendations

Based on the findings of the study, several actionable recommendations are proposed to assist police departments and policymakers in mitigating turnover intention and enhancing organizational effectiveness.

1. Enhance Organizational Culture: Reform efforts should focus on cultivating a supportive and inclusive culture, emphasizing ethical leadership, fairness in decision-making, and transparent communication.

2. Invest in Officer Well-being: Institutions should provide access to mental health services, stress management training, and wellness programs to reduce burnout and improve satisfaction.

3. Tailor Retention Strategies: Customize retention policies to address different career stages and roles. Younger officers may prioritize opportunities for advancement, while senior officers may value stability and recognition.

4.  Promote Leadership Development: Training programs focusing on emotional intelligence, ethical decision-making, and supportive supervision can strengthen leadership capacity and positively impact subordinate satisfaction and commitment.

5.  Conduct Culture Diagnostics: Regular assessments of organizational culture and climate can help detect misalignments between institutional values and officers' lived experiences, allowing for timely and targeted reforms.

6.  Encourage Feedback Mechanisms: Establish safe and anonymous channels through which officers can voice concerns and suggestions without fear of retaliation, fostering a culture of openness and continuous improvement.

7.  Develop structured peer support programs: Future initiatives could examine how peer support structures enhance psychological well-being, reduce occupational stress, and foster organizational commitment, especially in high-risk work environments like policing.

8.  Utilize digital tools for well-being monitoring: Leverage technology such as mobile applications to monitor real-time indicators of officer well-being and organizational satisfaction, providing valuable data for timely interventions.

9.  Integrate psychological resilience training: Programs designed to build officers resilience could enhance their coping capacity in high-stress situations, potentially reducing burnout and turnover intention. Future research could evaluate the effectiveness of such interventions in law enforcement settings.

10. Conduct cross-cultural comparative studies: Future studies could compare organizational culture, job satisfaction, and turnover intention among police officers across ASEAN countries, providing broader regional insights and policy learning opportunities.

11. Explore the impact of the physical work environment: In addition to psychosocial and cultural factors, research could assess how physical working conditions such as facilities, equipment, and workspace design influence job satisfaction and retention in policing.

## Conclusion

This study contributes to the growing body of research on employee retention by providing empirical insights into the relationships between organizational culture, job satisfaction, and turnover intention within the context of Malaysian policing. The findings affirm that both organizational culture and job satisfaction are significant predictors of turnover intention, highlighting their central roles in shaping officer's decisions to stay or leave the force. However, the moderating role of job satisfaction was not statistically significant, suggesting that job satisfaction alone may have limited capacity to buffer the influence of organizational culture on turnover intention in high-stress occupational environments such as law enforcement. These findings suggest that while fostering a strong organizational culture and increasing job satisfaction can each independently reduce officer's intentions to leave, efforts to align cultural values with officers expectations remain essential.

Notably, this study demonstrates that strong organizational cultures do not automatically translate into lower turnover intentions, particularly when those cultures are perceived as rigid, hierarchical, or lacking in fairness and support. Similarly, while job satisfaction remains a critical factor for workforce well-being, it may not fully compensate for negative organizational experiences or misaligned institutional values. For police departments and policymakers, these results highlight the need to prioritize initiatives that build supportive, cohesive organizational cultures and actively monitor factors contributing to job satisfaction, such as leadership support, meaningful work, and opportunities for advancement. Interventions should address not only organizational structures but also officer's perceptions of cultural alignment to enhance retention. Some related important focuses are on strengthening ethical leadership, promoting organizational justice, supporting officers' mental health, and creating inclusive environments where officers feel valued, supported, and aligned with organizational goals.

Future research should expand on these findings by employing longitudinal designs to examine causal pathways over time, incorporating qualitative methodologies to capture the lived realities of police officers, and testing alternative moderators such as perceived organizational support, occupational identity, psychological resilience, and leadership style. Such research will deepen understanding of the complex drivers of turnover intention and inform the development of comprehensive, evidence-based strategies to improve retention, engagement, and well-being within the Royal Malaysia Police and other similarly demanding professions.

Future research should also consider qualitative approaches to gain deeper insights into officer's experiences of organizational culture, job satisfaction, and value alignment. Exploring these issues through interviews or focus groups could help to identify additional contextual and personal factors influencing turnover intentions and further inform strategies to support workforce stability in policing.

**Matrix**

[DataSet1] D:\AMOS Group Data\Erny\Journal 2025\Data Journal.sav

**************** PROCESS Procedure for SPSS Version 4.2 ****************

Written by Andrew F. Hayes, Ph.D. www.afhayes.com

Documentation available in Hayes (2022). www.guilford.com/p/hayes3

**************************************************************************

Model: 1
Y: F_ILO
X: C_OCUL
W: G_JS
Sample
Size: 373

**************************************************************************

OUTCOME VARIABLE:
F_ILO
Model Summary
R R-sq MSE F df1 df2 p
.4814 .2318 1.4628 37.1098 3.0000 369.0000 .0000
Model
coeff se t p LLCI ULCI
constant −2.8346 2.2722 −1.2476 .2130 −7.3026 1.6334
C_BudOrg 1.1495 .4546 2.5286 .0119 .2556 2.0435
G_KepKer 1.0021 .4050 2.4744 .0138 .2057 1.7985
Int_1 -.1231 .0783 −1.5712 .1170 -.2771 .0310
Product terms key:
Int_1: C_OCUL x G_JS
Test(s) of highest order unconditional interaction(s):
R2-chng F df1 df2 p
X*W .0051 2.4687 1.0000 369.0000 .1170
----------
Focal predict: C_OCUL (X)
Mod var: G_JS (W)
Data for visualizing the conditional effect of the focal predictor:

Paste text below into a SPSS syntax window and execute to produce plot.

```
DATA LIST FREE/
C_OCUL G_JS F_ILO.
BEGIN DATA.
4.0410 4.7690 4.2177
5.0253 4.7690 4.7715
6.0096 4.7690 5.3252
4.0410 5.5777 4.6259
5.0253 5.5777 5.0817
6.0096 5.5777 5.5374
4.0410 6.3865 5.0341
5.0253 6.3865 5.3919
6.0096 6.3865 5.7497
END DATA.
GRAPH/SCATTERPLOT=
C_OCUL WITH F_ILO BY G_JS.
```

********************* ANALYSIS NOTES AND ERRORS ************************

Level of confidence for all confidence intervals in output:
95.0000

## Author contributions

**Conceptualization:** Erny Yusnida Che Yusoff.

**Supervision:** Mohd Nasir Selamat, Rusyda Helma Mohd.

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
