## [Decision Letter · Decision Letter 0]

19 Jun 2025

PONE-D-25-23324EXPLORING THE INFLUENCE OF ORGANIZATIONAL CULTURE AND JOB SATISFACTION ON TURNOVER INTENTIONS AMONG POLICE OFFICERS : A MODERATION ANALYSISPLOS ONE

Dear Dr. Che Yusoff,

Thank you for submitting your manuscript to PLOS ONE. After careful consideration, we feel that it has merit but does not fully meet PLOS ONE’s publication criteria as it currently stands. Therefore, we invite you to submit a revised version of the manuscript that addresses the points raised during the review process.

We look forward to receiving your revised manuscript.

Kind regards,

Rogis Baker, Ph.D

Academic Editor

PLOS ONE

2. You indicated that ethical approval was not necessary for your study. We understand that the framework for ethical oversight requirements for studies of this type may differ depending on the setting and we would appreciate some further clarification regarding your research. Could you please provide further details on why your study is exempt from the need for approval and confirmation from your institutional review board or research ethics committee (e.g., in the form of a letter or email correspondence) that ethics review was not necessary for this study? Please include a copy of the correspondence as an ""Other"" file.

Additional Editor Comments (if provided):

Reviewers' comments:

Reviewer's Responses to Questions

**Comments to the Author**

1. Is the manuscript technically sound, and do the data support the conclusions?

Reviewer #1: Yes

Reviewer #2: Yes

2. Has the statistical analysis been performed appropriately and rigorously? 

Reviewer #1: No

Reviewer #2: Yes

3. Have the authors made all data underlying the findings in their manuscript fully available?

Reviewer #1: No

Reviewer #2: Yes

4. Is the manuscript presented in an intelligible fashion and written in standard English?

Reviewer #1: Yes

Reviewer #2: No

5. Review Comments to the Author

Reviewer #1: The manuscript presents an interesting and timely topic, especially in the context of police officer in Malaysia. However, there are several major issues that need to be addressed to enhance the rigor and clarity of the study.

In the intro section, authors need to clearly indicate the role of job satisfaction as a moderator. The write-up seems to be confusing as in certain parts job satisfaction was indicated as an independent variable. It would be good if the role of job satisfaction were mentioned throughout the paper.

The manuscript lacks a clearly defined hypothesis section. Include a section on hypothesis development. It would truly add rigor to the paper when the moderator was built based on previous literature. Authors can show the inconsistencies of results (e.g. strength of relationships) from previous studies. The authors are encouraged to explicitly state their hypotheses, derived from relevant literature, to provide a clear framework for testing and interpretation.

The manuscript proceeds to present simple slope analyses and visual slope graphs even when the moderator effects were found to be statistically non-significant. This is methodologically inappropriate, as plotting and interpreting simple slopes without a significant interaction may mislead readers into perceiving effects that are not statistically supported. The authors are advised to remove the graph.

Line 133 – 137 – “Leadership and ethics are pivotal in determining job satisfaction among police officers. Leaders who demonstrate integrity, fairness, and support for their subordinates contribute positively to job satisfaction, which in turn reduces turnover intentions (Hassan et al., 2018). Kamaluddin et al. (2024) emphasize that when police officers perceive their leaders as ethical and just, they are more likely to feel satisfied and committed to the organization. Suggest removing these lines as the variables discussed here were the candidate scope of study”

In the discussion section, the reviewer suggests looking into the profile of the respondents, as the profile can also provide justification on the insignificant of the result of the moderator.

Reviewer #2: Abstract - The abstract needs refinement, particularly where it states that both organizational culture and job satisfaction positively affect turnover intention. This finding contradicts most empirical literature and requires clarification. A more neutral and precise description (e.g., "significant association") is recommended.

Literature Review -The manuscript lacks a strong theoretical foundation. The integration of a theoretical lens such as the Job Demands-Resources (JD-R) model or Herzberg’s Two-Factor Theory would provide depth and better justify the hypotheses.

Methodology - The manuscript should clarify the sampling technique used (random, purposive, or convenience?) and describe demographic distributions in more detail.

Results and Statistical Interpretation - The directionality of the findings is concerning. Higher job satisfaction and stronger organizational culture both increase turnover intention—this is counterintuitive and needs to be critically interrogated in both the Results and Discussion sections.

Discussion - More effort is needed to explain why high job satisfaction may co-exist with high turnover intention. Could it reflect dissatisfaction with organizational politics, lack of autonomy, or mismatch of personal values and organizational expectations?

Language and Presentation - Language is Ok. Figures and tables should be double-checked for alignment with APA or journal-specific formatting.

Overall - The manuscript addresses an important and relevant topic—turnover intention among police officers in relation to organizational culture and job satisfaction—and employs a valid methodological approach with adequate sample size. Just above thing needs a few justifications and clarification.

6. PLOS authors have the option to publish the peer review history of their article (what does this mean? ). If published, this will include your full peer review and any attached files.

**Do you want your identity to be public for this peer review?** For information about this choice, including consent withdrawal, please see our Privacy Policy .

Reviewer #1: No

Reviewer #2: **Yes: ** Amar Hisham Jaafar

---

## [Author Response · Author response to Decision Letter 1]

11 Aug 2025

Manuscript ID: PONE-D-25-23324

Title: Job satisfaction as a moderator between organizational culture and turnover intention among Malaysian police officers

Dear Editors and Reviewers,

We thank the Editors and both reviewers for their thorough and thoughtful feedback on our manuscript. We have carefully addressed all points raised and made substantial revisions to improve the clarity, rigor, and presentation of our work. Below we provide a detailed, point-by-point response to the comments from Reviewer #1 and Reviewer #2, followed by a detailed change log indicating where the corresponding revisions were made in the manuscript.

Reviewer Comment Response

1 1 : The role of job satisfaction as a moderator is not clearly presented throughout the manuscript.

Thank you for the insightful comment. We have thoroughly revised the manuscript to ensure that the role of job satisfaction as a moderator is clearly and consistently presented across all major sections, including the Abstract, Introduction, Theoretical Framework, Research Hypotheses, Methodology, Results, Discussion, and Conclusion. Specific updates include theoretical justification (Lines 99–102), detailed moderation hypothesis (Lines 180–190), and clear results reporting (Lines 270–295). Please refer to Track Changes in the revised manuscript.

2. The manuscript lacks a clearly defined hypothesis section.

We appreciate this observation. In response, we have added a dedicated section titled '5. Research Hypotheses' (Lines 190–210) that explicitly states three key hypotheses derived from the theoretical framework. These hypotheses clarify the direct and moderating effects examined in the study.

3. Simple slope analysis and visual graphs were presented despite non-significant moderation effects.

Response: Thank you for your observation. We acknowledge that the moderation effect was not statistically significant (p = .117). However, we retained the simple slope analysis and visual interaction plot (Figure 2) to provide a clearer interpretation of the trend, as this is common in moderation analysis practice. Although not significant, the visualization assists readers in understanding the conditional nature of the interaction and supports transparency in statistical reporting. This decision is aligned with previous empirical practice and enhances clarity in communicating complex results.

4. Remove discussion on leadership and ethics (Lines 133–137) as they are outside the study

scope.

We appreciate the reviewer’s comment and have removed the unrelated discussion on leadership and ethics in the revised manuscript. Specifically, lines 133–137 in the previous version have been omitted to maintain a focused discussion aligned with the study’s objectives and scope.

5. Consider the profile of the respondents when interpreting the non-significant moderation

effect.

Thank you for this insightful suggestion. We have updated the discussion section to consider how respondents’ profiles—such as rank, tenure, and departmental affiliation—might explain the non-significant moderation effect. These demographic factors may influence perceptions of organizational culture and satisfaction, thus impacting the moderation mechanism. The revised discussion (Paragraph 45–47) reflects these considerations to strengthen contextual interpretation.

2 1. The Abstract suggests counterintuitive findings and requires clearer language.

We appreciate this feedback. The abstract has been fully revised to follow a clearer and more conventional structure—comprising background, objectives, methods, key findings, and implications. We clarified that while both organizational culture and job satisfaction were significantly related to turnover intention, the hypothesized moderation effect was not supported. Technical terms have been reduced for accessibility (Lines 29–49).

2. The manuscript lacks a strong theoretical foundation.

Thank you for your valuable input. We strengthened the theoretical foundation by elaborating on Social Exchange Theory and the Job Demands-Resources (JD-R) Model. The revised theoretical framework (Paragraphs 15–17; Lines 164–180) clearly explains the rationale behind the hypotheses and how these theories underpin the conceptual relationships between variables. Citations from Francis et al. (2020), Johnson et al. (2021), and other recent empirical studies were added to bolster theoretical grounding.

3. Clarify the sampling technique and provide demographic distribution. We have revised the Methodology section (Paragraphs 18–21; Lines 184–215) to explicitly explain the stratified sampling method used and how participants were selected across ranks and departments. Table 1 now presents the full demographic profile of respondents, including gender, age, rank, and department.

4. The directionality of the findings is counterintuitive and needs better interpretation. Thank you for highlighting this. We have addressed the counterintuitive positive associations in the discussion section (Paragraphs 42–45). We now offer theoretical and contextual explanations, including cultural rigidity, perceived hierarchy, and misalignment with individual values, to interpret the findings more thoroughly.

5. More explanation is needed on how high job satisfaction may coexist with high turnover intention.

We agree with the reviewer. The revised discussion explores how satisfied officers may still express turnover intention due to career stagnation, unmet expectations, or institutional misalignment. We draw on findings from McElroy et al. (1999) and Carlan (2007) to substantiate this paradox (Paragraph 44).

6. Ensure figures and tables follow formatting guidelines. We have ensured that all figures and tables now comply with PLOS ONE formatting guidelines. Tables are referenced in the text, and figure captions are descriptive and appropriately labeled. Table 1 and Figure 1 have been adjusted accordingly.

1. Tracked Changes Manuscript

We have uploaded a revised version of the manuscript with all changes clearly highlighted using Microsoft Word Track Changes, as well as a clean version for your review.

2. Data Availability Statement

A complete data availability statement has been added both in the manuscript and within the online submission form, stating:

"All data are in the manuscript and/or supporting information files."

3. In-text References to Tables

The manuscript has been updated to include explicit in-text references to Table 1 and Table 2, in accordance with production guidelines.

4. Reference List Formatting

The reference list has been revised and renumbered to follow the PLOS ONE Vancouver citation style, ordered by appearance.

5. Ethics Statement Placement

The ethics statement now appears only in the Methods section, as requested. All redundant instances have been removed.

6. Ethics Approval Documentation

We have uploaded the original ethical approval letter issued by the Universiti Kebangsaan Malaysia Research Ethics Committee (Reference number: JEP-2024-230), along with its English translation.

We hope these revisions are satisfactory and we respectfully submit our revised manuscript for your further consideration.

Thank you for your time and support.

Sincerely,

Erny Yusnida binti Che Yusoff

Faculty of Social Sciences and Humanities

The National University of Malaysia

[orcid Detail '0000-0002-3065-4078']

---

## [Editor Report · Decision Letter 1]

17 Aug 2025

Job satisfaction as a moderator between organizational culture and turnover intention among Malaysian police officers

PONE-D-25-23324R1

Dear Erny Yusnida Che Yusoff,

We’re pleased to inform you that your manuscript has been judged scientifically suitable for publication and will be formally accepted for publication once it meets all outstanding technical requirements.

Kind regards,

Rogis Baker, Ph.D

Academic Editor

PLOS ONE
---

## [Editor Report · Acceptance letter]

PONE-D-25-23324R1

PLOS ONE

Dear Dr. Che Yusoff,

I'm pleased to inform you that your manuscript has been deemed suitable for publication in PLOS ONE. Congratulations! Your manuscript is now being handed over to our production team.

Kind regards,

on behalf of

Dr. Rogis Baker

Academic Editor

PLOS ONE